# Curriculum Fine-tuning of Vision Foundation Model for Medical Image Classification Under Label Noise

**Yeonguk Yu**    **Minhwan Ko**    **Sungho Shin**    **Kangmin Kim**    **Kyoobin Lee**[*]
Gwangju Institute of Science and Technology
{yeon_guk, mhko1998, hogili89, kgmin156}@gm.gist.ac.kr    kyoobinlee@gist.ac.kr

## Abstract

Deep neural networks have demonstrated remarkable performance in various vision tasks, but their success heavily depends on the quality of the training data. Noisy labels are a critical issue in medical datasets and can significantly degrade model performance. Previous clean sample selection methods have not utilized the well pre-trained features of vision foundation models (VFMs) and assumed that training begins from scratch. In this paper, we propose CUFIT, a curriculum fine-tuning paradigm of VFMs for medical image classification under label noise. Our method is motivated by the fact that linear probing of VFMs is relatively unaffected by noisy samples, as it does not update the feature extractor of the VFM, thus robustly classifying the training samples. Subsequently, curriculum fine-tuning of two adapters is conducted, starting with clean sample selection from the linear probing phase. Our experimental results demonstrate that CUFIT outperforms previous methods across various medical image benchmarks. Specifically, our method surpasses previous baselines by 5.0%, 2.1%, 4.6%, and 5.8% at a 40% noise rate on the HAM10000, APTOS-2019, BloodMnist, and OrgancMnist datasets, respectively. Furthermore, we provide extensive analyses to demonstrate the impact of our method on noisy label detection. For instance, our method shows higher label precision and recall compared to previous approaches. Our work highlights the potential of leveraging VFMs in medical image classification under challenging conditions of noisy labels.

## 1   Introduction

Deep neural networks have demonstrated remarkable performance across various tasks, including classification, detection, and segmentation [20, 16, 19, 57]. In medical imaging, these neural networks leverage large amounts of labeled data to train models capable of accurately detecting or classifying medical conditions from images such as dermatoscopes, X-rays, MRIs, and CT scans. However, in practical settings, data often contain noisy labels and it is well established that neural networks perform well only when the quality of training data is sufficiently high [42, 29, 5]. Noisy labels occur when the data annotations—the labels assigned to training images—are incorrect or inconsistent. This issue is particularly problematic in medical imaging, where annotating images is more complex compared to natural images [53, 25]. Consequently, improving the robustness of neural networks against noisy labels is a crucial area of research, directly affecting the effectiveness and reliability of medical imaging technologies.

A large number of algorithms have been developed to address the issue of performance degradation caused by noisy samples [42]. In particular, clean sample selection methods, such as MentorNet [24], Co-teaching [17], Co-teaching+ [56], JoCor [47], and CoDis [50], have demonstrated superior performance without requiring modifications to the model architecture or training loss. The core

---

[*]Corresponding author: Kyoobin Lee. Our code is available at github.com/gist-ailab/CUFIT.

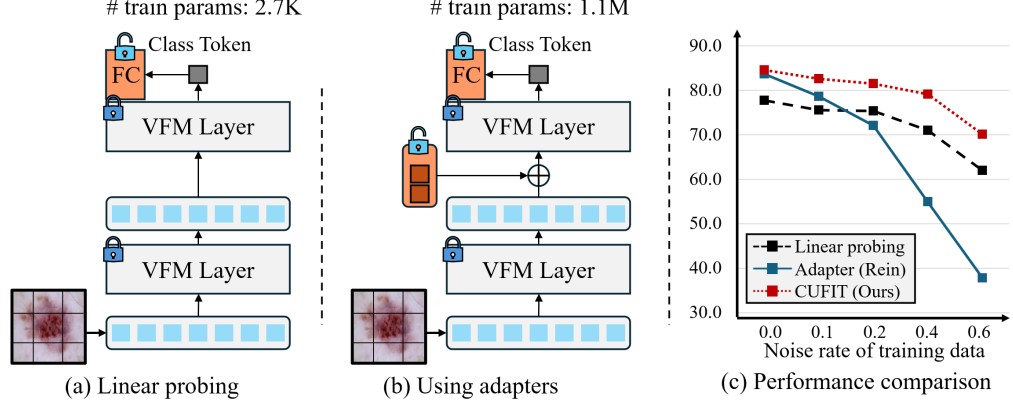

<p style="text-align:center">(a) Linear probing       (b) Using adapters       (c) Performance comparison</p>

Figure 1: Illustration of linear probing (a) and adapter usage (b). Specifically, the weights of the foundation model are frozen, while the fully connected layer or adapter weights (shown in orange) are updated during the training phase. In (c), a performance comparison using a simulated noisy dataset (HAM10000) is presented. It demonstrates that linear probing is more robust to noisy labels compared to the adapter, whereas the adapter outperforms linear probing when there are no noisy labels.

principle behind these methods is that small-loss samples are likely to be clean, as they are easier to classify and the model memorizes them faster. Additionally, using two different but homogeneous networks to select small-loss samples for each other is more stable than relying on a single model for sample selection. These methods have shown outstanding performance using traditional neural network architectures based on convolutional layers. However, there are practical issues with these methods in two aspects: (i) it cannot be guaranteed that these methods will perform equally well with transformer-based architectures, which have recently gained significant attention, and (ii) their assumption that training starts from scratch is impractical, as it prevents the use of rich features from pre-trained models, which could be beneficial for filtering noisy labels.

Recently, large-scale vision foundation models (VFMs) with transformer-based architectures [13], such as CLIP [39], MAE [18], SAM [28], and DINOv2 [37], have gained attention for their performance and applicability across various tasks. The self-supervised training of VFMs on large-scale datasets enhances their robustness against various image corruptions and improves their generalization capabilities [10, 48, 38, 40]. The inherent robustness and rich features of VFMs can be beneficial for detecting noisy labels. For instance, linear probing of VFMs is relatively unaffected by noisy samples since it does not modify the VFM's feature extractor, preventing the memorization of noisy data, as shown in Figure 1. However, linear probing does not fully leverage the VFM's capabilities when there is a domain gap between the pretraining task of the VFM and the target task (e.g., pretraining on natural images versus medical image classification) [48]. To address this issue, some researchers have proposed using trainable fine-tuning adapters for VFMs [22, 23, 48, 9]. Yet, these adapters might degrade performance by memorizing noisy labels due to their trainable parameters involved in feature extraction. Therefore, we state our research question as follows: *How can we use the power of pre-trained vision foundation models for medical image classification in the presence of noisy labels?*

In this paper, we introduce a **Cu**rriculum **Fi**ne-**T**uning paradigm for vision foundation models in medical image classification under noisy labels, called **CUFIT**. CUFIT is a curriculum-learning framework designed to fine-tune VFMs with noisy medical datasets. The framework consists of three training modules: the Linear Probing Module (LPM), the Intermediate Adapter Module (IAM), and the Last Adapter Module (LAM). During the training stage, clean samples are selected based on an *agreement* criterion, where a sample is selected if its annotation matches the module's prediction. Specifically, the LPM is trained using all available samples, as linear probing is robust against noisy labels. Subsequently, the IAM is trained with the samples selected by the LPM, and the LAM is trained with the samples selected by the IAM. This inter-module curriculum training (i.e., LPM→IAM→LAM) is beneficial for increasing the number of clean samples available to train the LAM, considering that the LPM only selects a limited number of samples due to performance degradation caused by the domain gap. Consequently, CUFIT leverages the LAM for final predictions,

offering strong fine-tuning performance for medical image classification in the presence of noisy labels by utilizing the strengths of both linear probing and adapters, as illustrated in Figure 1-c. In summary, the main contributions of this paper are as follows:

- We introduce CUFIT, a simple yet effective fine-tuning paradigm for medical image classification using VFMs in the presence of noisy labels. This method leverages the robustness of linear probing and the generalization capability of fine-tuning adapters to handle noisy datasets during training stage.

- We conduct various experiments demonstrating that CUFIT significantly improves the robustness of VFMs against noisy labels in medical datasets. We show that CUFIT outperforms previous methods across several medical image benchmarks.

- We provide extensive analyses to enhance the understanding of our fine-tuning paradigm. Additionally, we validate our framework with various VFMs and adapter configurations.

## 2 Related Work

**Vision foundation models.**   Vision transformers (ViTs) embed 2D images into 1D tokens and model their global correlations using the self-attention mechanism [13, 35, 43]. ViTs are known to be effective when large datasets are used, and the concept of vision foundation models is introduced. Several studies have developed pre-trained vision transformers based on self-supervised learning. For instance, contrastive language-image pre-training (CLIP) provides high-quality visual representations through contrastive learning with a large amount of image-text pairs [39]. Additionally, masked auto-encoder (MAE) offers high-capacity models that generalize well through self-supervised learning with masked auto-encoding, where the task is to reconstruct token patches from the given masked tokens [18]. Moreover, knowledge distillation with no labels (DINOv1) [7] proposed a teacher-student framework for self-training without annotations, resulting in a well-generalized ViT. More recently, DINOv2 introduced a self-training framework that combines masked autoencoding and teacher-student training based on carefully curated datasets [37].

Since large models and self-training in VFMs provide strong generalization capabilities for various tasks, parameter-efficient fine-tuning has gained attention. Parameter-efficient fine-tuning (PEFT) aims to adapt foundation models to new tasks by training only a few adapter parameters while keeping the model itself frozen. Notably, there is research that proposes low-rank adaptation (LoRA), which introduces trainable rank decomposition matrices into each layer of the transformer architecture in large language models [22]. For the VFMs, visual prompt tuning [23] proposed appending prompts to the input sequence of each transformer block, achieving excellent fine-tuning performance with minimal parameters. Similarly, Adaptformer [9] introduces a novel MLP block to replace the original one in transformer blocks, allowing for the use of both original and few trainable parameters. More recently, Wei *et al.* proposed Rein, which aims to adapt VFMs for semantic segmentation with domain generalization capabilities [48]. Also, Dutt *et al.* investigate PEFT algorithms across both convolutional and large transformer-based networks for medical image classification, demonstrating the effectiveness of PEFT, particularly in the low-data regimes common in medical imaging [14]. In this paper, we focus on fine-tuning VFMs for image classification in the presence of noisy labels using adapters. Rather than introducing a new adapter, we utilize an existing adapter within our training paradigm.

**Learning with noisy label.**   Deep neural networks have demonstrated remarkable performance on large-scale datasets. However, it is well-known that neural networks can easily memorize noisy labeled samples, leading to degraded performance. Several studies have been conducted to explore robust supervised learning in the presence of noisy labels. These studies can be categorized into five approaches [42]: (i) robust architectures, (ii) robust regularization, (iii) robust loss functions [15, 33, 46, 59], (iv) loss adjustment [26, 31, 32], and (v) sample selection [24, 36, 17, 56, 50]. In this paper, we categorize our method as a sample selection method, which selects samples with clean labels from a noisy training dataset. While previous sample selection methods typically consider training from scratch, we focus on training starting from a pre-trained model, which is known to be more robust to noisy labels [21]. Additionally, research has explored using CLIP to enable robust training by leveraging its text-image matching capability on noisy datasets [49].

Various methods have been proposed for clean sample selection from noisy datasets. For example, MentorNet introduced the use of a teacher network to guide the student network to focus on clean labels [24]. Similarly, Decoupling proposed updating two networks by using only the samples with differing predictions between them [36]. Co-teaching also trains two networks simultaneously, updating them based on sample recommendations from each other [17]. Co-teaching+ [56] improved upon Co-teaching by introducing the "update by disagreement" strategy, where only the samples with differing predictions between the two networks are used. More recently, Xia *et al.* proposed CoDis, an extension of Co-teaching+ that employs an "update by discrepancy" strategy, selecting samples with high-discrepancy prediction probabilities between the two networks to utilize more samples [50]. These methods are based on the assumption that clean samples can be identified using certain criteria, and that network collaboration is more stable than self-selection, which may lead to error accumulation. In this paper, we develop our method based on same assumption, but we assume that we start the learning process from the pre-trained VFM.

## 3 Problem Setup

We consider a $k$-class classification task using a neural network. Let $\mathcal{X} \in \mathbb{R}^d$ denote the input space and $\mathcal{Y} \in \mathbb{R}^k$ represent the ground-truth label space. In a typical classification task, the neural network is trained to align the input space with the label space. To this end, a training dataset $D = \{(x_i, \hat{y}_i)\}_{i=1}^n$ is used for supervised learning with cross-entropy loss. In practice, a sample $(x_i, \hat{y}_i)$ is considered as a noisy labeled sample when human-annotated label $\hat{y}_i$ does not match the true label $y_i$. The objective of this paper is to develop a fine-tuning approach for VFMs that is robust to noise and capable of performing accurately on noisy datasets.

Given a pre-trained VFM with parameters $\theta_{\text{VFM}}$, consisting of a sequence of layers (e.g., attention blocks in ViT [13]) $L_1, L_2, ..., L_M$, where $M$ is the depth of $\theta_{\text{VFM}}$, the learning objective for a classification problem can be formulated as:

$$\min_{\theta_t} \sum_{i=1}^{n} \mathcal{L}_{ce}(p(x_i|\theta_{\text{VFM}}, \theta_t), \hat{y}_i), \tag{1}$$

where $\theta_t$ and $\mathcal{L}_{ce}$ represent the parameters targeted for updating and the cross-entropy loss, respectively. Here, $p(\cdot|\theta)$ refers to the prediction for a given input using parameters $\theta$. We refer to the training process as linear probing when $\theta_t$ is limited to the parameters of the linear layer $\theta_l$. Additionally, we refer to the training process as full-tuning when $\theta_t$ includes $\theta_{\text{VFM}}$, and as adapter tuning when it includes adapter parameters $\theta_a$, which are not part of $\theta_{\text{VFM}}$.

## 4 Method

In this section, we begin by describing the adapter method for fine-tuning the VFM in Section 4.1. Following this, in Section 4.2, we introduce our method, CUFIT, which utilizes three modules: a linear layer and two adapters, to combat noisy labels. The key idea behind CUFIT is to leverage the well-pre-trained features of the VFM without updating the feature extractor when handling corrupted samples. Subsequently, the adapters are trained using the samples selected in a curriculum-based training manner, as shown in Figure 2 (i.e., linear probing → intermediate adapter → last adapter). This approach helps increase the number of selected samples by reducing the domain gap between the pretraining task and the medical image task. It is important to note that our framework does not train the modules sequentially (i.e., where one module starts training only after another finishes); instead, it trains the modules simultaneously on the current batch, similar to multi-task training.

### 4.1 Learning with Adapter

We consider various adapters for fine-tuning VFMs on medical image datasets.In particular, adapters like Visual Prompt Tuning (VPT [23]), AdaptFormer [9], Low Rank Adaptation (LoRA [22]) and Rein [48] can be used. These methods have been shown to be efficient for various image and video tasks, even compared to full model training [9, 48]. Typically, when an adapter is used for fine-tuning, the parameters of the VFM are frozen and not included in the optimization process.

In this section, we briefly introduce how an adapter works. Note that our goal is not to propose a novel adapter but rather to present a training paradigm that can be applied to various adapters. For

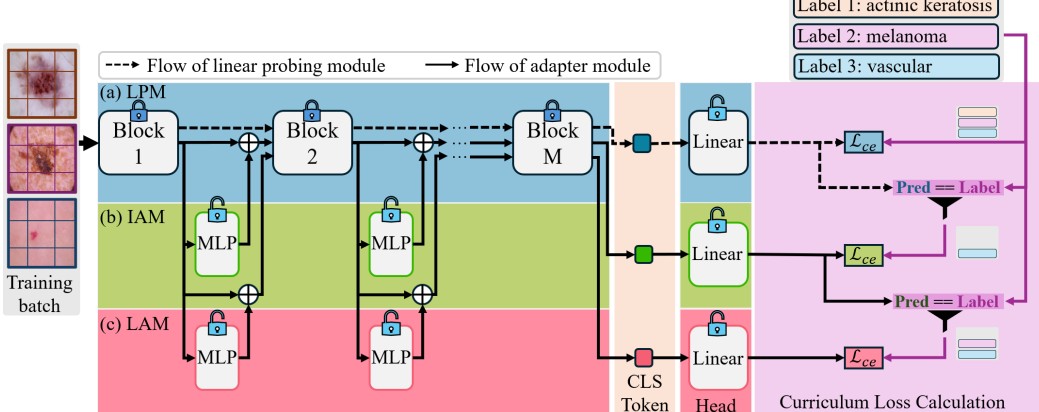

Figure 2: Illustration of our proposed training framework, CUFIT, which consists of a pre-trained VFM and three distinct modules: (a) the linear probing module (LPM), (b) the intermediate adapter module (IAM), and (c) the last adapter module (LAM). During the training stage, the LPM selects clean samples for the IAM based on the *agreement* criterion, and the IAM selects clean samples for the LPM. During the inference stage, only the LAM is used for prediction.

vision transformers (ViTs), the output of the attention block for the given input patches is calculated as follows:

$$x_l' = \text{Attention}(Q, K, V) = \text{Softmax}(\frac{QK^T}{\sqrt{d}}V) + x_{l-1}, \tag{2}$$

where $x_{l-1}$ is the output token of the previous block. Here, $Q$, $K$, and $V$ refer to the query, key, and value vectors, respectively, which are derived from linear projection and LayerNorm [6] applied to $x_{l-1}$. The final output of the block, $x_l$, is then computed using LayerNorm and an MLP. Without using an adapter, this process is formulated as:

$$x_l = \text{MLP}(\text{LN}(x_l')) + x_l', \tag{3}$$

where $x_l$ is the output token of the $l$-th block. When an adapter is used, the Eq 3 is replaced by the following:

$$x_l = \text{MLP}(\text{LN}(x_l')) + x_l' + \text{Adapt}(x_{l-1}; \theta_{l-1}^a), \tag{4}$$

where $\text{Adapt}(\cdot; \theta_l^a)$ refers to the adapter function for the $l$-th layer, parameterized by $\theta_l^a$. We consider this process to be an arbitrary function, as various adapters can be used. In the last block, the [CLS] token is passed to the following linear layer for final image classification.

## 4.2 Curriculum training of three different modules

We consider a pre-trained VFM, $\theta_{\text{VFM}}$, with a single linear layer parameterized by $\theta_{\text{LPM}} \in \mathbb{R}^{c \times k}$, an intermediate adapter module parameterized by $\theta_{\text{IAM}}$, and a last adapter module parameterized by $\theta_{\text{LAM}}$, where $c$ refers to the dimension of the class token (e.g., 384 dimensions for the ViT-small architecture). Then, we propose a curriculum training framework for these three modules, in which the LPM is trained with all samples from the given batch, while the adapter modules are trained with filtered samples selected by their corresponding module using the *agreement* criterion. The *agreement* criterion refers to a method where a sample is considered clean if the module's prediction matches the sample's annotation. The idea behind this criterion is that a robust classifier will correctly predict the sample under the assumption that clean labels are in the majority within a noisy class. Therefore, a sample is selected as clean if it meets the *agreement* criterion (e.g., a "dog" image with a "dog" annotation). Thus, we build the curriculum training framework based on the robustness of the LPM against noisy labels using the *agreement* criterion.

In particular, the linear probing module (LPM) is trained as follows:

$$\min_{\theta_{\text{LPM}}} \sum_{i=1}^n \mathcal{L}_{ce}(p(x_i|\theta_{\text{VFM}}, \theta_{\text{LPM}}), \hat{y}_i), \tag{5}$$

which directly represents supervised learning using the given images and corresponding labels. Here, $p(x_i|\theta_{\text{VFM}}, \theta_{\text{LPM}})$ refers to the output of the network using $\theta_{\text{VFM}}$ and $\theta_{\text{LPM}}$ for the given image $x_i$. During the training stage, the intermediate adapter module (IAM) is trained as follows:

$$\min_{\theta_{\text{IAM}}} \sum_{i=1}^{n} \mathcal{L}_{ce}(p(x_i|\theta_{\text{VFM}}, \theta_{\text{IAM}}), \hat{y}_i) \mathbb{1}\{\arg\max p(x_i|\theta_{\text{VFM}}, \theta_{\text{LPM}}) = \hat{y}_i\}, \tag{6}$$

where $\mathbb{1}\{\cdot\}$ is the indicator function. This simple modification using the indicator function ensures that the adapter module is trained only on selected samples chosen by the linear layer. Finally, the last adapter module (LAM) is trained as follows:

$$\min_{\theta_{\text{LAM}}} \sum_{i=1}^{n} \mathcal{L}_{ce}(p(x_i|\theta_{\text{VFM}}, \theta_{\text{LAM}}), \hat{y}_i) \mathbb{1}\{\arg\max p(x_i|\theta_{\text{VFM}}, \theta_{\text{IAM}}) = \hat{y}_i\}. \tag{7}$$

The Eq 7 is equivalent to Eq 6, but it uses LAM and IAM instead of IAM and LPM, respectively. This simple yet effective sample selection strategy is well-suited for fine-tuning the VFM on noisy image datasets. Notably, it does not require any hyperparameters like the estimated noise rate, which are commonly needed in previous works [17, 47, 50], where they assume the noise rate is known in order to select small-loss samples (e.g., selecting 60% of samples in a batch for a known noise rate of 40%). After training is completed, only the last adapter module is used to predict the given test image.

## 5 Experiments

### 5.1 Settings

**Datasets.** We evaluate our approach on four simulated noisy label medical multi-class image classification benchmarks: HAM10000 [44], APTOS-2019 [11], BloodMnist [3], and OrgancMnist [52]. Additionally, we conduct an evaluation on a real-world noisy label benchmark, Kaggle-EyePACS [2]. In particular, the detail of datasets are as follows:

- HAM10000 [44]: This dataset contains 10,015 dermatoscopic images for skin lesion classification, with each image classified into one of seven possible disease categories. We use all the images for training, and the evaluation is conducted using the 1,512 test images provided by the ISIC 2018 challenge [1].

- APTOS-2019 [11]: This dataset consists of 3,662 retina images taken with fundus photography under various imaging conditions. Each image is rated for the severity of diabetic retinopathy (DR) on a scale from 0 to 4. We use 2,930 images for training and 366 images for evaluation.

- BloodMnist [3]: This dataset contains 17,092 images of individual cells, with each image annotated as one of eight possible cell types. We use 11,959 images for training and 3,421 images for evaluation.

- OrgancMnist [52]: This dataset includes 23,538 images that are center-sliced from the Hounsfield-Unit of 3D images in a coronal view. Each image is labeled as one of eleven body organs. We use 12,975 images for training and 8,216 images for evaluation.

- Kaggle-EyePACS [2]: This Kaggle competition dataset provides 35,126 retina images categorized into five DR severity grades for training, which are known to contain noisy labels [25]. Specifically, some DR category labels (e.g., mild DR labeled as moderate DR) are noisy, and some images considered normal may actually contain retinal diseases such as glaucoma or drusen, which are not included in the classification categories. It is estimated that there is approximately a 30%–40% label error in this dataset [25, 45]. We use the original 35,126 training images and their annotations for training, and all images from APTOS-2019 for evaluation. Additionally, we use the FGADR [60] dataset for further evaluation.

**Baselines.** We compare the performance of CUFIT with basic training paradigms: full training, linear probing, and fine-tuning with Rein [48]. Additionally, we evaluate our approach against other training-based methods, including Co-teaching [17], JoCor [47], and CoDis [50]. Like ours, these methods do not modify the training loss or architecture. Specifically, Co-teaching trains two networks simultaneously, with each network selecting small-loss samples from its peer's predictions to guide

| Dataset | Noise rate | Method | | | | | | |
|---|---|---|---|---|---|---|---|---|
| | | Full-training | Linear probing | Rein | Co-teaching | JoCor | CoDis | CUFIT |
| HAM10000 | 0.1 | 66.5 | 75.6 | 78.6 | 81.5 | 81.1 | 81.9 | **82.6** |
| | 0.2 | 62.6 | 75.3 | 72.1 | 79.1 | 79.4 | 80.1 | **81.5** |
| | 0.4 | 56.1 | 71.0 | 54.9 | 74.3 | 73.9 | 74.1 | **79.1** |
| | 0.6 | 59.9 | 61.9 | 37.8 | 67.3 | 67.1 | 66.1 | **70.1** |
| | Mean | 61.3 | 71.0 | 60.8 | 75.5 | 75.4 | 75.5 | **78.3** |
| APTOS-2019 | 0.1 | 66.8 | 79.2 | 82.5 | 82.8 | **84.8** | 83.2 | 84.2 |
| | 0.2 | 65.9 | 79.4 | 78.7 | 81.2 | 83.1 | 82.0 | **84.2** |
| | 0.4 | 69.9 | 79.5 | 77.2 | 79.5 | 76.0 | 79.5 | **81.6** |
| | 0.6 | 48.2 | 66.9 | 42.0 | 72.9 | 74.2 | 75.7 | **76.3** |
| | Mean | 62.7 | 76.3 | 68.9 | 79.1 | 79.5 | 80.1 | **81.6** |
| BloodMnist | 0.1 | 95.4 | 97.2 | 95.9 | 98.6 | 98.5 | 98.5 | **99.0** |
| | 0.2 | 93.9 | 96.7 | 89.0 | 97.6 | 97.3 | 97.2 | **98.8** |
| | 0.4 | 91.8 | 95.8 | 69.3 | 93.7 | 93.0 | 93.5 | **98.3** |
| | 0.6 | 87.9 | 90.3 | 45.6 | 88.7 | 87.3 | 88.0 | **98.2** |
| | Mean | 92.3 | 95.0 | 75.0 | 94.7 | 94.0 | 94.3 | **98.6** |
| OrgancMnist | 0.1 | 85.3 | 83.3 | 87.4 | 92.1 | 92.1 | 92.1 | **93.7** |
| | 0.2 | 79.9 | 82.9 | 82.0 | 90.9 | 91.9 | 90.7 | **93.6** |
| | 0.4 | 72.1 | 79.9 | 63.8 | 85.8 | 85.3 | 85.8 | **91.6** |
| | 0.6 | 64.5 | 72.2 | 43.1 | 82.8 | 82.6 | 81.9 | **87.4** |
| | Mean | 75.5 | 79.6 | 69.1 | 87.9 | 88.0 | 87.6 | **91.6** |

Table 1: Average test accuracy (%) on four simulated noisy datasets with different noise levels. The test accuracy is averaged over the last ten epochs. The best and second-best results in each case are highlighted in **bold** and underline, respectively.

| Testset | Method | | | | | | |
|---|---|---|---|---|---|---|---|
| | Full-training | Linear probing | Rein | Co-teaching | JoCor | CoDis | CUFIT |
| APTOS-2019 | 34.2 | 65.4 | 69.1 | **70.9** | 69.3 | 69.2 | 69.8 |
| FGADR | 14.3 | 46.4 | 48.8 | 44.9 | 53.1 | 53.0 | **53.7** |
| Total | 27.5 | 59.0 | 62.3 | 62.2 | 63.9 | 63.8 | **64.4** |

Table 2: Average test accuracy (%) on real-world noisy datasets (Kaggle-EyePACS for training). After the training is done, we evaluate the model on two datasets: APTOS-2019 and FGADR. The best result and second-best result in each case are highlighted in **bold** and underline, respectively.

learning. JoCor extends this idea by incorporating co-regularization to maximize agreement between the two networks. CoDis further refines this process by selecting samples that not only have small losses but also show high divergence between the two networks. It is important to note that we do not compare our proposed framework with state-of-the-art methods that modify the training loss (e.g., semi-supervised learning) or model architecture [51, 8]. In the experiments, we apply these methods to VFMs with adapters, as they do not require specific model architectures, and VFMs with adapters outperform the linear probing of VFMs (i.e., DINOv2 with the Rein adapter is used as the default setting for training with Co-teaching, JoCor, and CoDis for a fair comparison).

**Implementation details.** For the experiments, we use DINOv2 [37] with the ViT-small [13] backbone as our basic vision foundation model. Additionally, we use Rein [48] as the fine-tuning adapter, originally proposed for domain-generalized semantic segmentation of VFMs. In our setup, we utilize the class token from the block for classification, rather than the patch tokens from multiple blocks.

We use the PyTorch [4] codebase for our experiments. BloodMnist and OrgancMnist datasets are sourced from MedMnist [54, 55]. We use the ViT-small architecture and the Adam optimizer [27]. All training runs for 100 epochs with a batch size of 32. The initial learning rate is 0.001, which decays by a factor of 10 at epochs 50, 75, and 90. For full-parameter training, however, we start with an initial learning rate of 0.0001. For the simulated noisy label benchmarks, we generate symmetric noise [17] for evaluation, with noise rates set at 10%, 20%, 40%, and 60%.

## 5.2 Simulated noisy medical image classification benchmark

First, we evaluate our framework on simulated noisy label benchmarks using four medical datasets. The average classification test accuracy for each dataset is provided in Table 1. Our framework consistently outperforms previous baselines, demonstrating its effectiveness under noisy labels by leveraging the pre-trained features of DINOv2 and the Rein adapter. Notably, our framework proves

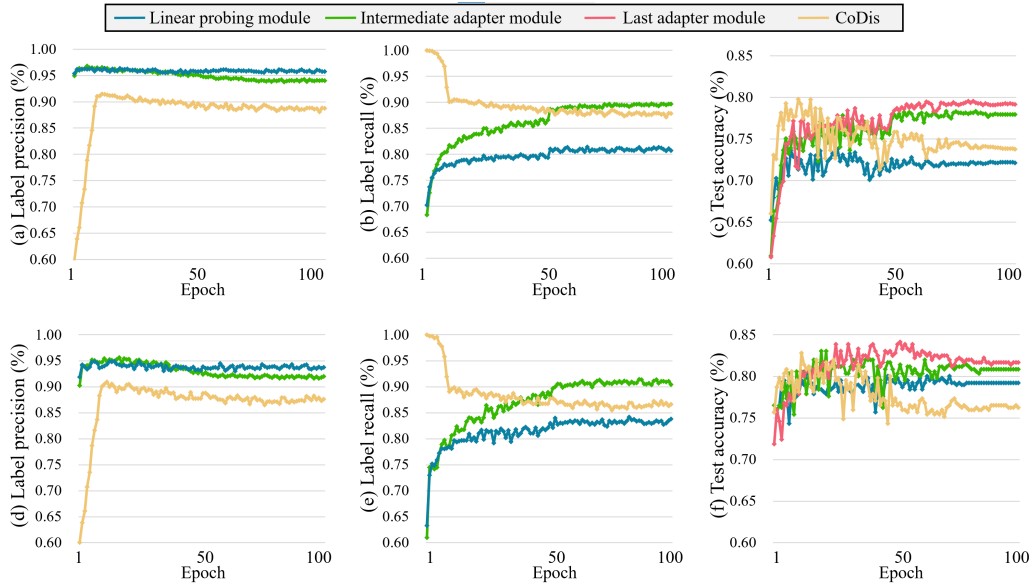

Figure 3: Illustration of label precision (a,d), label recall (b,e), and test accuracy (c,f) vs. epoch. The first row is for HAM10000 with 40% noise rate, and the second row is for APTOS-2019 with 40% noise rate.

to be more effective as the noise rate increases. For example, CUFIT achieves 0.85% relateively higher accuracy than CoDis on HAM10000 with a 10% noise rate, while the improvement rises to 3.7% at a 60% noise rate. This result indicates that the pre-trained features of the VFM are particularly useful for handling noisy labels in the given datasets.

### 5.3 Real-world noisy medical image classification benchmark

We train DINOv2 with Rein on the real-world benchmark, using the Kaggle-EyePACS dataset for training and the APTOS-2019 and FGADR datasets for testing. Given the highly imbalanced training set (e.g., approximately 73% of the samples are labeled as the normal class), we use weighted cross-entropy loss to train the model. Since previous sample selection methods require a noise rate hyperparameter, we employed the noise estimation method from [34], following Co-teaching [17].

In Table 2, we report the classification accuracy on the APTOS-2019 and FGADR datasets, as well as the overall accuracy across both datasets. Our method outperforms other baselines on the FGADR dataset and the combined dataset, while Co-teaching achieves the highest accuracy on the APTOS-2019 dataset. We believe this discrepancy is due to the distribution of normal class samples—approximately 50% in APTOS-2019 and about 5% in FGADR. Co-teaching performs well in classifying the normal class, whereas our method excels at classifying diseased samples. For example, our method achieves 53.9% macro-average test accuracy, while Co-teaching achieves 48.5% on the combined test set.

## 6 Discussion

### 6.1 How does CUFIT works?

So far, we have demonstrated through empirical results that our framework significantly improves the robustness of VFM fine-tuning against noisy labels. However, we have not yet discussed why our framework is effective in learning with noisy labels. In Figure 3, we present label precision, label recall, and test accuracy over the number of epochs to illustrate how our framework functions. In principle, higher label precision indicates fewer noisy samples in the selected data, while higher label recall indicates fewer clean samples in the unselected data. We have following three observations:

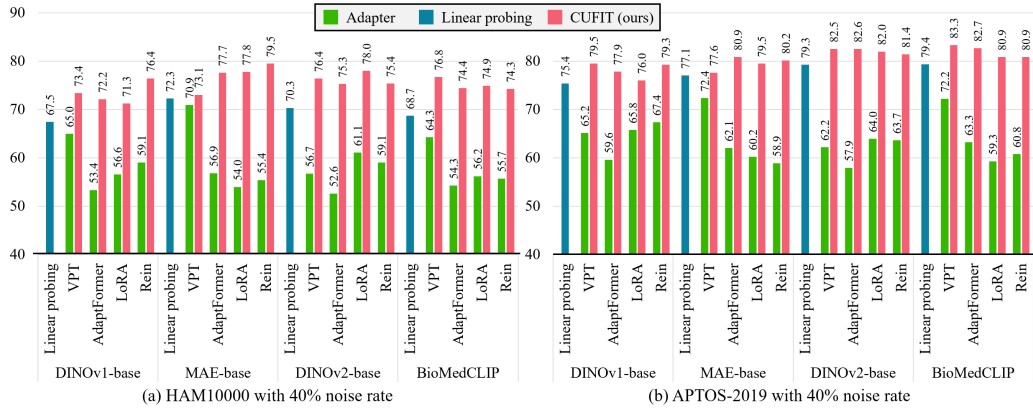

(a) HAM10000 with 40% noise rate    (b) APTOS-2019 with 40% noise rate

Figure 4: Test accuracy of our method with various VFMs (DINOv1 [7], MAE [18], DINOv2 [37]) and adapters (VPT [23], AdaptFormer [9], Rein [48]). We use HAM10000 and APTOS-2019 with 40% noise rate for training.

| Dataset | Noise Rate | Full-training | | Linear probing | | CoDis | | CUFIT | | |
|---|---|---|---|---|---|---|---|---|---|---|
| | | ResNet | DINOv2 | ResNet | DINOv2 | ResNet | DINOv2 | ResNet | ResNet+rein | DINOv2 |
| HAM10000 | 0.2 | 73.1 | 66.5 | 71.1 | 75.6 | 74.9 | 80.1 | 77.7 | 79.9 | 82.6 |
| | 0.4 | 59.6 | 62.6 | 67.8 | 75.3 | 72.4 | 74.1 | 73.8 | 75.4 | 81.5 |
| APTOS-2019 | 0.2 | 80.4 | 66.8 | 80.3 | 79.2 | 80.3 | 82.0 | 82.4 | 82.2 | 84.2 |
| | 0.4 | 64.7 | 65.9 | 73.4 | 79.4 | 78.1 | 79.5 | 80.5 | 82.2 | 84.2 |

Table 3: Average test accuracy on simulated noisy datasets (HAM10000 and APTOS-2019) using the ResNet50 architecture. Test accuracy is averaged over the last ten epochs.

- We find that CoDis exhibits lower label precision during training compared to LPM and IAM, suggesting that previous sample selection methods fail to effectively utilize the pre-trained features of VFMs, leading to lower test accuracy. These methods train the network on all training data without sample selection during the early stages of training (e.g., epochs 1 to 10 in their default setting). However, this approach may harm the feature extraction capability of VFMs and result in degraded performance.

- LPM consistently achieves the highest label precision across epochs but has the lowest label recall, indicating that it effectively prevents the memorization of noisy samples. However, because the feature extractor remains unchanged, its overall accuracy is limited, thus selecting only a small number of clean samples.

- IAM, on the other hand, achieves similar label precision but higher label recall by leveraging the adapter module, which contributes to the improved test accuracy of LAM. This suggests that by adapting the feature extractor through training the adapter on a few certain clean samples, IAM can be a more accurate module. It can then provide more clean samples to LAM, resulting in better overall performance.

### 6.2 Performance comparison across various VFMs and adapters

To validate the performance of CUFIT in various settings, we present experimental results using three VFMs and adapters in Figure 4. We use the same experimental setup (i.e., 100 epochs with the Adam optimizer) to train the network across all backbones and adapters. Specifically, we utilize four backbones, including DINOv1 [7], MAE [18], DINOv2 [37], and BioMedCLIP [58] and four adapters, including VPT [23], AdaptFormer [9], LoRA [22], and Rein [48]. BioMedClip is a CLIP-like model trained with the PMC-15M dataset, which contains 15 million biomedical image-text pairs collected from 4.4 million scientific articles. Our results demonstrate that our framework consistently helps build a robust classifier across different VFMs and adapters. For example, our framework achieves better performance compared to both adapter-based methods and linear probing. Additionally, we observe that linear probing consistently outperforms the adapter method in all cases, indicating that the performance of adapters can be degraded by noisy labels across various adapters.

| Dataset | Noise rate | Method | | | | | | |
|---------|-----------|--------------|----------------|------|-------------|-------|-------|-------|
|         |           | Full-training | Linear probing | Rein | Co-teaching | JoCor | CoDis | CUFIT |
| CIFAR10 | 0.8 | 25.9 | 79.0 | 24.8 | 78.2 | 75.7 | 76.3 | **83.9** |
| CIFAR100 |  | 6.3 | 59.6 | 25.6 | 66.7 | 64.2 | 63.7 | **73.8** |
| ANIMAL10N | 0.08* | 74.5 | 89.1 | 88.0 | 92.2 | 91.9 | 91.7 | **92.3** |

Table 4: Average test accuracy on the natural image dataset with simulated noisy labels (CIFAR, symmetric noise at 80%) and real-world noisy labels (ANIMAL10N [41], which has an estimated noise ratio of 8%). The test accuracy is averaged over the last ten epochs. We use DINOv2 with Rein adapter for the experiment. **Bold** values the best result.

## 6.3 Performance on CNNs with adapters

We designed CUFIT for VFMs due to their strong pre-trained feature extraction capabilities, enabled by self-supervised training on large datasets. However, CNN-based architectures like ResNet [20] also utilize pre-trained weights instead of starting from scratch. Therefore, we validate CUFIT on ImageNet [12] pre-trained ResNet50, with and without the Rein adapter modified for ResNet. The experimental results are shown in Table 3. We observe that our method outperforms other training paradigms when using the ImageNet pre-trained ResNet architecture. Additionally, the Rein adapter for ResNet improves performance, demonstrating that using fewer trainable parameters with an adapter, compared to full training, is beneficial for combating noisy labels. Finally, we show that the more representative pre-trained features of DINOv2 outperform the ImageNet pre-trained features across all training methods.

## 6.4 Performance on noisy natural image classification benchmark

Since our framework is easily applicable not only to medical image classification but also to natural image classification, we present experimental results on the CIFAR [30] simulated noisy classification benchmarks and ANIMAL10N [41] real-world noisy classification benchmark in Table 4. We validate our framework under an extremely high noise rate setting (80%) for CIFAR benchmark, as it is intuitive that our framework performs well under low noise rates due to the feature extraction capabilities of VFM. As shown in Table 4, our framework outperforms other sample selection methods in natural image classification benchmarks as well. This demonstrates the effectiveness of our framework, highlighting that using well pre-trained VFMs is beneficial for detecting noisy labels in natural images, as expected.

## 7 Conclusion

This paper presents a curriculum fine-tuning paradigm called CUFIT, designed to robustly fine-tune vision foundation models (VFMs) for medical image classification. Our framework is based on the insight that linear probing of VFMs is robust to noisy labels, as it does not modify the feature extraction process. Building on this, CUFIT consists of three training modules: the linear probing module (LPM), the intermediate adapter module (IAM), and the last adapter module (LAM). These modules are trained simultaneously, with each selecting clean samples for the next module. Specifically, while the LPM is trained on all samples, the LPM and IAM select clean samples for the IAM and LAM, respectively (i.e., LPM→IAM→LAM). Experiments demonstrate that CUFIT significantly improves the performance of VFMs in the presence of noisy labels for medical image classification. Additionally, we provide extensive analyses to enhance the understanding of CUFIT. We hope our insights inspire future research to further explore the robustness of vision foundation models when learning with noisy labels for various medical imaging tasks.

## Acknowledgments

This work was partly supported by Institute of Information & communications Technology Planning & Evaluation (IITP) grant funded by the Korea government(MSIT) (No. RS-2022-II0951, Development of Uncertainty-Aware Agents Learning by Asking Questions, 90%) and Institute of Civil Military Technology Cooperation funded by the Defense Acquisition Program Administration and Ministry of Trade, Industry and Energy of Korean government under grant No. 22-CM-GU-08, 10%).

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
