# OpenReview forum: "Curriculum Fine-tuning of Vision Foundation Model for Medical Image Classification Under Label Noise"
_NeurIPS.cc/2024/Conference — NeurIPS 2024 poster_

### Official Review · Reviewer_tyjY · 2024-06-17

**Soundness:** 2
**Presentation:** 2
**Contribution:** 2
**Rating:** 4
**Confidence:** 4

**Summary:**

The authors propose Cufit, a curriculum fine-tuning method for improving the performance of medical image classification under the noisy labels setting. The method shows strong performance against other baselines on several medical datasets. The authors have also provided results on a non-medical dataset.

**Strengths:**

**Strengths**

1. The proposed strategy Cufit outperforms other baselines included in the evaluation
2. The strategy is agnostic to models (CNNs and ViTs) and images (medical and natural)
3. Cufit does not require knowledge about certain hyperparameters which are required in other methods.
4. The authors have presented results on non-medical images as well.

**Weaknesses:**

**Weaknesses**

1. The paper is very poorly written with spelling and grammatical errors throughout the text. Certain elements in the text are written in a convoluted way that confuses the reader.
2. The entire paper is about the proposed method “Cufit” but no section in the paper describes the algorithm in detail. Section 6.1 (“How does Cufit work?”) does not talk about the method but only about the results.
3. It is important to cite previous work on PEFT in medical image analysis such as [1] in the text.
4. The paper lacks an explanation of the baseline methods used in the evaluation. Methods like CoDis are briefly mentioned in the Related Work section but have not been defined anywhere else. People unfamiliar would not be able to understand Cufit and how it differs from previously proposed methods.
5. The experiments should include the CheXPert dataset. Apart from being frequently adopted for medical image analysis problems, it would also evaluate the proposed methodology for multi-label classification problems. Furthermore, CheXPert is supposed to contain noisy labels due to automatic labelling from free-text reports. Hence, it would adequately test the proposed method Cufit under a noisy label setting.
6. To make the experiments more extensive, more datasets and PEFT methods (see [1] for reference) should be included. LoRA is one of the most popular PEFT strategies used for transformer-based models (especially in the case of medical image classification [1]) and should be included in the experiments.
7. For natural image classification, the authors have adopted the CIFAR dataset. Firstly, in order to provide conclusive results, several natural imaging datasets should be included. Secondly, there are many datasets much more appropriate than CIFAR that should have been used instead.

References
1. Dutt, Raman, et al. "Parameter-Efficient Fine-Tuning for Medical Image Analysis: The Missed Opportunity." *Medical Imaging with Deep Learning*, 2024, https://openreview.net/forum?id=LVRhXa0q5r.

**Questions:**

**Questions**

1. After reading the paper several times, I am yet to understand how Cufit actually works. There is no section dedicated in the text that explains this.

2. Why are methods like LoRA that are very frequently used for doing PEFT in transformer-based models and datasets like CheXpert excluded from the analysis?

Please address the **Weaknesses** section for more questions.

**Limitations:**

Please address the **Weaknesses** section.

---

> ### Author Rebuttal · Authors · 2024-08-04
>
> ### **Q2, W6:  Experimental results with LoRA**
> We greatly appreciate your suggestion. We have conducted the experiment primarily using the Rein adapter, as it achieves excellent performance in domain-generalized semantic segmentation, and we believe it has chance to have excellent performance in the medical domain. Given that LoRA is a well-known PEFT method, we agree it should be included. We will add the experimental results using LoRA in Section 6.2, 'Performance comparison across various VFMs and adapters.' The following table provides the comparison results with LoRA on APTOS-2019 and HAM10000 with noise-rate of 40%, showing that our method is also applicable with LoRA. We will include this result in Section 6.2.
> | Dataset | Full-training | Linear probing | LoRA | Co-teaching | JoCor | CoDis | Ours |
> | --- | --- | --- | --- | --- | --- | --- | --- |
> | APTOS-2019 | 69.9 | 79.5 | 63.9 | 79.5 | 80.1 | 80.1 | **81.7** |
> | HAM10000 | 56.1 | 71.0 | 57.0 | 78.1 | 77.2 | 75.0 | **78.8** |
>
> ### **W3: Previous work on PEFT in medical image analysis.**
> Thank you for pointing out the related work that we missed. We will cite the paper [1] in the related work section to address the importance of PEFT in medical image analysis.
>
> ### **Q2, W5: Experiment with CheXPert.**
> Thank you for your valuable insight and feedback. In our current version of the paper, we focused on methods for multi-class classification problems in the medical domain, which is why we excluded datasets for multi-label classification problems. However, we agree with your suggestion regarding the CheXPert dataset. It is frequently used in medical image analysis and contains noisy labels due to automatic labeling from free-text reports. Incorporating CheXPert to evaluate Cufit in multi-label classification problems will be a part of our future research.
>
> ### **W4: Other dataset for natural image classification.**
> To validate our method on natural image classification, we further evaluate using the Animal10N dataset, a real-world noisy dataset. We follow the experimental settings outlined in "SELFIE: Refurbishing Unclean Samples for Robust Deep Learning" (i.e., batch size of 128 and 100 epochs training). Also, for baseline methods, we set the noise rate of 0.08 which is the estimated noise rate by "SELFIE" paper. Given the small performance gap, we conduct the experiment three times and average the results. The following table provides the experimental results using DINOv2-small with the Rein adapter. We will add these results to the paper.
>
> | Dataset | Full-training | Linear probing | Rein | Co-teaching | JoCor | CoDis | Ours |
> | --- | --- | --- | --- | --- | --- | --- | --- |
> | Animal10N | 74.5 | 89.1 | 88.0 | 92.2 | 91.9 | 91.7 | **92.3** |
>
> ### **W7: Detailed explanation of the baseline methods.**
> We agree on your opinion that our paper lacks an explanation of the baselines. We further describe details of the other methods after line 230. For example, the following paragraph will be added in the text.
> - Previous method such as Co-teaching, JoCor, and CoDis involve training two neural networks concurrently on the dataset. In the training stage, each network identifies and discards potentially mislabeled examples, subsequently teaching the other network with the remaining clean examples. Specifically, Co-teaching trains two homogeneous networks simultaneously and cross-updates parameters using R(T)% small-loss instances from the given mini-batch (R(T)% is the estimated noise rate). Additionally, JoCor encourages two different networks to make predictions closer to each other by explicit regularization loss during training to select the small-loss sample with assumption that two networks easily reach an agreement on its predictions for clean samples. CoDis selects high-discrepancy examples with a joint loss (joint loss is equal to cross-entropy loss reduced by Jensen–Shannon (JS) divergence loss), leveraging discrepancy for sample selection, thereby improving selection quality.
> ### **Q1 W2: Detailed explanation of our method**
> We appreciate your feedback. To address lack of detailed explanation of our method, we will update the Section 6.1. and Section 4. to provide description about the method. Our method is designed to train the VFMs with a adapter module robustly for noisy medical image datasets. Our key insight is as follows.
> - Cufit leverages a curriculum learning approach to robustly fine-tune the VFMs for noisy datasets with three key modules: Linear Probing Module (LPM), Intermediate Adapter Module (IAM), and Last Adapter Module (LAM).
> - Linear Probing Module (LPM): The LPM is first trained on all available samples in the given batch. Linear probing is relatively robust against noisy labels as it does not modify the feature extraction capability of VFM, thus preventing memorization of noisy samples. However, since there is no modification in feature extractor, its accuracy is low.
> - Intermediate Adapter Module (IAM): For the given batch, the IAM is trained using samples selected by the LPM based on agreement criteria, which considers a sample as clean if its annotation matches the prediction of the LPM. Compared to LPM, IAM can achieve higher accuracy since it has small PEFT on the feature extraction.
> - Last Adapter Module (LAM): Similarly, the LAM is trained with samples selected by the IAM. This inter-module curriculum training (LPM → IAM → LAM) increases the number of clean samples available for training the LAM, enhancing its performance. Specifically, LPM only selects few samples since its accuracy is lower than IAM. LPM is used for the inference stage.
> ### **W1: Spelling and grammatical errors throughout the text.**
> We apologize for the poor quality of current version of our paper. We have conducted a thorough review of the paper to ensure it has no spelling and grammatical errors throughout the text. Also, we plan to have the professionally English proofreading service to further refine the language and clarity.

---

> > ### Comment · Reviewer_tyjY · 2024-08-11
> > **Thanks for the Rebuttal**
> >
> > Thanks for the Rebuttal. My questions have been answered to some extend and I have revised my rating.

---

> > > ### Author Response · Authors · 2024-08-12
> > > **Thank you for comments!**
> > >
> > > Dear Reviewer tyjY,
> > >
> > > Thanks for raising the score rating. We really appreciate your feedback to improve our manuscript.
> > >
> > > Best regards,
> > >
> > > The Authors

---

> ### Comment · Area_Chair_LurH · 2024-08-10
> **Check rebuttal**
>
> The authors have provided feedback to review comments. Please take time to read the rebuttal and provide response. Thank you.
>
> AC

---

### Official Review · Reviewer_T7UR · 2024-07-05

**Soundness:** 3
**Presentation:** 2
**Contribution:** 3
**Rating:** 6
**Confidence:** 5

**Summary:**

In this paper, the authors propose a curriculum learning strategy for fine-tuning on noisy medical datasets. The key insight comes from that linear probing with limited training samples can be more robust to label noise. The performance is good compared to the former methods.

**Strengths:**

Generally, I think this is a good paper.  Noisy label is a severe and practical problem for medical scenarios due to diagnosis uncertainty and the intuition behind the proposed method is clearly stated. The performance looks good and the proposed method is clean and efficient.

**Weaknesses:**

However, I still have to point out some details are not clearly demonstrated. Please check the questions for more details. I will change my score based on the authors' responses.

**Questions:**

1. Figure 2 is confusing. The core content in the method may fall in the Curriculum order and the agreement criteria. But for now, it is not clearly reflected in the figure which makes section 4.2. I suggest the author reorganize the figure and detail the purple block.

2. I am not very clear about how the mode is trained. Commonly, when referring to curriculum learning, I guess the model will be trained with multiple steps but in equation 6 authors also modify the loss function which makes it also like a multi-task training pipeline.  This confuses me a lot and the authors have to clarify it more clearly.

3. For the noise simulation. The authors have to state the details more, e.g., how are the correct labels changed just randomly or with some rules? and for multi-label or multi-class medical classification tasks, are there any differences? Similarly, this also confuses me in the method part. If a case with both pneumonia and bone fracture labels but the pneumonia part is wrong, will the whole case be dismissed for the latter training?

4. From lines 203 to 221, in introducing the dataset, it is important to specifically point out what datasets are multi-label tasks and what are multi-class, since in medical these two types all wide exist.

5. In section 6.2, it is also encouraging to explore more  performance differences between general VFM and medical-wise VFM like PMC-CLIP and BiomedCLIP. While considering the rebuttal time limitation, this is just a bonus suggestion.

6. A minor suggestion, I suggest the authors clarify the definition of noise in their introduction part more formally as though for ML domain noise is clear, in medical domain, the noise types are complex which may be hard to understand for the audience in clinical background.

**Limitations:**

The authors adequately addressed the limitations.

---

> ### Author Rebuttal · Authors · 2024-08-04
>
> ### **Q1, Q2: Clear description of our method.**
> We apologize for the unclear term usage. Our method trains the model like a multi-task training approach (i.e., all modules are trained simultaneously for a current given batch). We use the term “curriculum” to represent the order of the agreement criteria (i.e., the sample selection order is LPM -> IAM -> LAM). We will either change the term “curriculum” or include the sentence, “Our method trains modules simultaneously for the current given batch like a multi-task training,” in the method section. Moreover, we will update Figure 2 to clearly show the curriculum order of the training and the agreement criteria in a given training batch. The updated figure is included in the rebuttal PDF.
>
> ### **Q3: Description of label noise simulation.**
> We utilize symmetric noise generation for the experiment. For example, when the noise rate is 0.4 and the number of classes is 6, a sample assigned to class 1 has its annotation changed to one of the other classes with equal probability of 0.08. We will add a detailed description on lines 239-240 (implementation details) and a description of multi-class problem in the preliminary section. Also, we conducted the experiment with multi-class datasets only and will focus on the multi-label classification problem in the future.
>
> ### **Q4: Clear dataset description.**
> Thank you for your In the current version of paper, we only evaluate the method on multi-class dataset. Thus, All dataset introduced in the paper are multi-class dataset. We will clearly mention it in lines 203 to 221. Also, it will be a future research to apply our method for multi-label classification problems.
>
> ### **Q5: Performance using medical-wise VFM.**
> Thank you for your insightful suggestion that using medical-domain VFM like PMC-CLIP and BiomedCLIP. We ran the experiments using the image encoder of BiomedCLIP with ViT-base architecture following setting of section 6.2 (i.e., 100 epochs training with Adam optimizer). We find a similar trend in medical-wise VFM, where linear probing has better accuracy than adapter training when there are noisy labels on datasets. The following table demonstrates the experimental results (test accuracy) on BiomedCLIP for HAM10000 and APTOS-2019 with noise rate of 0.4 and we will add the results on Section 6.2.
>
> | dataset | Linear probing |  Rein | Ours |
> |:----------------:|:-------------:|:-----:|:----:|
> |  HAM10000  |     68.9 |     55.6 |  **74.4** |
> |  APTOS-2019  |     78.6 |      64.2   | **80.0** |
>
> ### **Q6: Definition of the label noise in the Introduction part.**
> We are thankful for your suggestion. We will add the sentence about definition of noise label in the Introduction section to help understanding the audience in non-ML domain. For example, line 25-26 “Noisy labels occur when the data annotations—the labels assigned to training images—are incorrect or inconsistent” can be replaced as “Noisy labels occur when the data annotations—the labels assigned to training images—are incorrect or inconsistent. In practice, the sample $(x_i, \hat{y}_i)$ is considered as a noisy labeled sample when human-labeled annotation $\hat{y}_i$ is not match the true label $y_i$”.

---

> ### Comment · Reviewer_T7UR · 2024-08-08
>
> Thanks a lot for the timely detailed rebuttals. I appreciate the authors' clarifications. Most of my concerns have been solved. My only left concern falls in the multi-class settings, as in healthcare, most cases are multi-label cases since patients may have different diseases, which will somewhat limit the paper's contributions and interests. Thus, sorry that I cannot increase my score more, while I clearly support this paper is worth a weak acceptance.

---

> ### Author Response · Authors · 2024-08-08
> **Thank you for all comments!**
>
> Dear Reviewer T7UR
>
> We would like to thank you again for all your constructive feedback. We will update our final version accordingly. Additionally, we will continue our work on multi-label cases as future work.
>
> Best regards,
>
> The Authors

---

### Official Review · Reviewer_1c7f · 2024-07-11

**Soundness:** 3
**Presentation:** 3
**Contribution:** 3
**Rating:** 7
**Confidence:** 4

**Summary:**

This paper presents a curriculum fine-tuning paradigm called Cufit. This method is designed to fine-tune Vision Foundation Models (VFMs) for medical image classification tasks under the presence of noisy labels. The approach leverages the robustness of linear probing and the generalization capabilities of fine-tuning adapters to improve model performance.

**Strengths:**

- Cufit is technically sound and is well-validated through extensive experimental results.
- The paper is well-structured and effectively conveys the motivation, approach, and outcomes.
- Demonstrated significant improvements in medical image classification performance under label noise.
- Applicability to both medical and natural image classification enhances the relevance of the framework.

**Weaknesses:**

- The training process seems to be complex and computationally intensive.
- I have concerns about the scalability of the proposed method. It may not scale well for very large datasets or in resource-constrained environments.

**Questions:**

- How does the computational cost of Cufit compare to other noise-robust training methods?

**Limitations:**

None.

---

> ### Author Rebuttal · Authors · 2024-08-02
>
> ### **Q1. Computational cost of Cufit compare to other noise-robust training methods.**
> We appreciate your feedback. Since the audience may be curious about the computational cost of our method and other training methods, we will provide the resource usage of these methods in the supplementary material. It may seem that our method is complex and computationally intensive. However, it utilizes resources reasonably compared to other methods. For example, we ran experiments on the HAM10000 dataset with a batch size of 32 using an RTX4090 GPU (the number of training images is 10,015) and found that our method performs effectively. The following table provides the memory usage and time cost for various methods with DINOv2-small architecture and Rein adapter.
>
> |                  | Full-training | Linear probing |  Rein | Co-teaching | JoCor | CoDis | Ours |
> |:----------------:|:-------------:|:--------------:|:-----:|:-----------:|:-----:|:-----:|:----:|
> |    Memory (MB)   |     4,350     |       824      | 3,144 |    10,086   | 5,542 | 10,836 | 5,604 |
> | Time (sec/epoch) |      18.6     |       6.6      |  16.5 |     32.9    |  32.7 |  51.4 | 37.7 |

---

> ### Comment · Area_Chair_LurH · 2024-08-10
> **Check rebuttal**
>
> The authors have provided feedback to review comments. Please take time to read the rebuttal and provide response. Thank you.
>
> AC

---

> > ### Comment · Reviewer_1c7f · 2024-08-10
> >
> > I have read the rebuttal and other reviews. I have no more questions. I would like to maintain my initial assessment.

---

> ### Author Response · Authors · 2024-08-11
> **Thank you for comments!**
>
> Dear Reviewer 1c7f,
>
> We appreciate your efforts in the reviewing process and your positive feedback.
>
> Best regards,
>
> The Authors

---

### Official Review · Reviewer_8Ng4 · 2024-07-13

**Soundness:** 3
**Presentation:** 3
**Contribution:** 3
**Rating:** 6
**Confidence:** 3

**Summary:**

The paper presents Cufit, a curriculum fine-tuning paradigm for Vision Foundation Models (VFM) aimed at improving medical image classification under label noise. This method leverages the robust feature extraction capabilities of pre-trained VFMs and employs a linear probing strategy to mitigate the impact of noisy labels. The curriculum fine-tuning process then utilizes clean sample selection to enhance the classification performance. The experimental results demonstrate that Cufit outperforms existing methods on various medical image benchmarks, showing significant improvements in classification accuracy.

**Strengths:**

1. The presentation is good.

**Weaknesses:**

1. The paper includes experimental comparisons with methods like JoCor and CoDis, but the discussion about these methods' performance is insufficient. The authors should provide a more detailed analysis of why JoCor and CoDis do not perform as well as Cufit. Understanding the strengths and weaknesses of these methods in comparison to Cufit would offer valuable insights.

2. The paper should further discuss the impact of noisy labels on different types of biomedical images. For some image types, noise may be less detrimental, while for others, it could significantly affect diagnostic accuracy. A more detailed exploration of how noise impacts various biomedical image datasets would enhance the comprehensiveness of the study.

**Questions:**

1. The paper includes experimental comparisons with methods like JoCor and CoDis, but the discussion about these methods' performance is insufficient. The authors should provide a more detailed analysis of why JoCor and CoDis do not perform as well as Cufit. Understanding the strengths and weaknesses of these methods in comparison to Cufit would offer valuable insights.

2. The paper should further discuss the impact of noisy labels on different types of biomedical images. For some image types, noise may be less detrimental, while for others, it could significantly affect diagnostic accuracy. A more detailed exploration of how noise impacts various biomedical image datasets would enhance the comprehensiveness of the study.

---

> ### Author Rebuttal · Authors · 2024-08-04
>
> ### **Q1: Discussion of previous methods**
> We greatly appreciate your valuable feedback about including strengths and weaknesses of previous methods in the paper. Our method outperforms previous methods in the medical image classification using VFMs. As shown in Figure 3 in the paper, our modules provide the higher label precision for selecting clean samples, which is based on the linear probing’s robustness for noisy labels. However, previous methods do not leverage the linear probing’s robustness, which leads to lower label precision. The followings are strengths and weaknesses of previous methods we argue.
> - (Strengths) Although previous methods do not leverage the linear probing, they can be applicable to any network. However, our method requires well pre-trained network such as DINOv2 or CLIP to start the training which is our main weakness and their strength against ours.
> - (Weaknesss) Previous methods do not leverage the linear probing’s robustness, which leads to lower label precision. Since they design the method based on the assumption that network is training from the scratch, their methods train the network for all training data without sample selection during early training stage (e.g.,  1 to 10 epoch in their default setting). However, this training for all data may harm the feature extraction ability of VFMs and lead to degraded performance. For instance, the Co-teaching method can have better performance using different sample selection start epoch for APTOS-2019 with 40% noise rate as shown in the following table. This demonstrates that the previous methods have varying performance according to the hyper-parameter and can have the best performance when the sample selection is started at right time (i.e., when a VFM with adapter memorizes clean samples but not yet memorizes noisy samples).
>
> | Sample selection start epoch  | 1 | 3 |  5 | 10 | 25 |
> |:----------------:|:-------------:|:--------------:|:-----:|:-----------:|:-----:|
> |   Test accuracy   |     78.9 |       **80.8** | 80.1 |  79.5 | 76.2 |
>
> We will include the analysis on the previous methods in the Discussion section and add limitation of our method in the Conclusion.
>
> ### **Q2: Discussion of the impact of noisy labels on different types of dataset**
> Thank you for your insightful feedback to improve our paper. Our main observation from experiments on the HAM10000 dataset (dermatoscopic images for skin lesion diagnosis) and the APTOS-2019 dataset (fundus photographs for grading diabetic retinopathy from “0-level no DR” to “4-level proliferative DR”) is that noisy labels harm APTOS-2019 less than they do HAM10000. Specifically, the relative performance drops for HAM10000 and APTOS-2019 are 34.3% and 7.1%, respectively (e.g., a 40% noise rate causes a 34.3% performance drop). For example, DINOv2s with Rein achieves 83.6% test accuracy with no noise and 54.9% test accuracy with 40% noise. We believe the relatively non-detrimental effect of noisy labels on APTOS-2019 arises from its continuous disease severity annotations. We will add this discussion to the Experiments section.

---

> ### Comment · Area_Chair_LurH · 2024-08-10
> **Check rebuttal**
>
> The authors have provided feedback to review comments. Please take time to read the rebuttal and provide response. Thank you.
>
> AC

---

### Author Rebuttal · Authors · 2024-08-04

We appreciate the reviewers for their valuable comments and constructive feedback on our paper. As summarized by all reviewers, we propose a novel parameter-efficient fine-tuning (PEFT) framework for medical image classification under noisy labels. We believe our framework can outperform previous noise-robust training methods, which focused on CNN networks trained from scratch.

This rebuttal addresses all reviewers’ concerns. Additionally, we provide a PDF file showing the modified version of Figure 2. Furthermore, we are open to discussions and committed to addressing any additional concerns from the reviewers to ensure the refinement of our paper.

---

### Decision · Program_Chairs · 2024-09-25

**Decision:**

Accept (poster)

**Comment:**

This paper studies vision foundation models for medical image classification from the aspect of parameter-efficient fine-tuning under noisy labels. The topic is interesting and valuable. Reviewers are generally positive for their comments. The strengths of the work include sound method design and rich experiments showing good performance. The weaknesses are mainly concerning lack of detailed analysis on comparison with SOTA, and clarifications on method/experiment implementation details. The rebuttal has satisfactorily answered most of the questions. With the final ratings and replies from reviewers, the AC would like to recommend acceptance for this submission.